# Not All Emotional Demands Are the Same: Emotional Demands from Clients’ or Co-Workers’ Relations Have Different Associations with Well-Being in Service Workers

**DOI:** 10.3390/ijerph17217738

**Published:** 2020-10-22

**Authors:** Joana Duarte, Hanne Berthelsen, Mikaela Owen

**Affiliations:** 1Centre for Work Life and Evaluation Studies (CTA), Malmö University, 21119 Malmö, Sweden; 2Centre for Work Life and Evaluation Studies (CTA) & the Faculty of Odontology, Malmö University, 21119 Malmö, Sweden; hanne.berthelsen@mau.se; 3Centre for Workplace Excellence, University of South Australia, Adelaide, SA 5001, Australia; Mikaela.Owen@unisa.edu.au

**Keywords:** emotional demands, emotional exhaustion, engagement, psychosocial safety climate, structural equation modelling, service work

## Abstract

There has been an increased interest in the study of emotional demands (ED) at work and its impact on workers’ well-being. However, ED have been conceptualized as a unitary concept, focused on interactions with clients, and excluding other potential sources of ED at work. Therefore, the aim of the current study is to explore the relation between ED from different relational sources (clients/patients/customers and colleagues, supervisors, and employees) and service workers’ exhaustion and engagement. Cross-sectional data from a sample of 2742 service workers were analysed using structural equation modelling. Results showed that ED from both sources (clients and colleagues) were associated with more emotional exhaustion, particularly if dealing with clients was not an integrated part of the role. Further, ED from clients’ relations were negatively associated with engagement for managers with staff responsibility, but positively for managers without staff responsibility. We also found moderating effects of psychosocial safety climate (PSC), whereby ED had the strongest effect on emotional exhaustion when PSC was low. This study suggests that different relational sources of ED at work have a different impact on employees’ well-being. Strategies that promote a reduction of extra-role ED, and the development of a PSC in the organization, could therefore offer possible solutions to promote employees’ psychological well-being and motivation.

## 1. Introduction

In the past couple of decades, there has been a shift away from an economy dominated by industrialization to an economy that is heavily reliant upon the service sector. This can be easily seen in Europe, with approximately 70 per cent of the workforce consisting of service workers, and more specifically 80 per cent in Sweden, an increase of 10 per cent in the last 20 years [1]. The now dominant service sector poses risks to workers in the form of emotional demands as a result of the interpersonal nature underpinning this line of work [2,3]. This has stimulated a wave of interest in the literature on the importance of emotional demands. However, despite the wealth of research into the health eroding consequences of emotional demands [4,5,6,7,8,9], as well as the de-motivational consequences under conditions of low resources [5,10,11], there is still limited understanding surrounding emotional demands in the workplace. Emotional demands have been understood as those aspects of a job that require sustained emotional effort due to interactions with clients [12]. However, this definition fails to consider interactions with fellow employees that can also require sustained emotional effort. As a result, the literature has yet to recognise the source of the emotional demands, and the potential differing impact of the different sources (e.g., clients and work relations). Therefore, the aim of the current study is to explore the impact of emotional demands from clients/patients/customers and emotional demands from work relations (e.g., colleagues, supervisors, and employees) onto workers’ exhaustion and engagement.

### 1.1. Consequences of Emotional Job Demands

With the crucial role the service sector, and more specifically service activities, is playing in the current economic climate, there is a recognition these activities go hand in hand with high emotional demands. For example, evidence indicates that occupations largely underpinned by end-user (e.g., clients, patients, or customers) interactions are exposed to the highest levels of emotional demands [3,13]. Emotional demands in this context are thought to stem from exposure to client suffering [9,14], client initiated violence [15,16], or high demands and expectations from clients [17]. These experiences and requirements can negatively impact the health and wellbeing of the service sector workers. Studies demonstrate that emotional demands are at least as important, or more important, than psychological-quantitative demands (e.g., workload) in terms of their impact on employee well-being in service work [9].

A large body of empirical evidence has shown that emotional job demands are related to a variety of poor health symptoms, particularly burnout/emotional exhaustion, e.g., [5,7,8,9,14], for a review). Emotional job demands have also been linked to other measures of poor health, such as depression [2,18], psychological distress [4], anxiety [19], and long-term sickness absence [6].

This relationship between emotional demands and employee poor health may be understood through the lens of the health erosion pathway of the job demands-resources model [20]. In accordance with the health erosion pathway, demanding conditions require effort and energy investment that may drain employees’ physical and psychological resources and lead to psychological strain (e.g., burnout). As such, exposure to high levels of emotional demands in the workplace is leading to a variety of detrimental health conditions.

Notwithstanding the large amount of support for a detrimental impact of emotional demands onto workers’ health, there are inconsistencies in the research. For example, there is evidence to indicate that emotional demands do not play a role in workers’ experience of health symptoms [21,22,23]. However, it should be noted that this evidence on the lack of association between emotional demands and health is largely outdated, as the majority of this research was conducted over 20 years ago, which calls into question the relevance of the findings to workers these days. Notwithstanding the potential lack of relevance, it does call into question the circumstances under which emotional demands have a detrimental impact on health.

Consistent with the findings from emotional demands to workers’ health and wellbeing, the association between emotional demands and organizational and individual work-related wellbeing outcomes is not clear cut. While emotional demands directly and negatively impact upon work-related wellbeing, such as low engagement and low job satisfaction [24,25], emotional demands do not occur in isolation. Within the working environment, workers also have access to resources. When workers have access to high levels of individual and/or job-related resources, high levels of emotional demands correspond positively with engagement [11,26], organizational connectedness [27], employee creativity, and work motivation [5]. That is, under emotionally demanding conditions workers notice and/or sufficiently utilize the motivational psychosocial factors in the workplace, job resources, to promote beneficial outcomes for both themselves and the organization.

### 1.2. Job Resources and Emotional Demands

As previously established, the beneficial link between job demands (e.g., emotional demands) and motivational outcomes (e.g., work engagement), often is only present under conditions of high job resources [11,26]. Job resources can also be beneficial in reducing the detrimental impact of job demands upon workers’ health [5,28,29].

In the extant literature, the focus has near exclusively targeted personal resources, such as self-efficacy or optimism, or task-level resources, such as autonomy or feedback. In the current research we will incorporate an organizational level resource, psychosocial safety climate, which extends beyond the narrow focus of a singular personal or task-level resource, to incorporate a multi-faceted resource that can address the multiple needs of the worker.

#### Psychosocial Safety Climate

Psychosocial safety climate (PSC) reflects the enacted policies, practices, and procedures as developed by senior management that address the protection of workers’ psychological health and safety above productivity goals [30]. It is an important organizational resource as it acts as a precursor to and moderator of psychosocial risk factors (i.e., job resources and job demands) in the workplace. Organizations that possess high PSC, as demonstrated through the priority given to the protection of workers’ psychological health and safety, provide increased availability and access to job resources, and reduced exposure to hazardous levels of job demands [30,31] as perceived by the workers. As a moderator, PSC reduces the negative impact of the psychosocial risk factor—job demand—on workers’ psychosocial and emotional health, and it enhances the positive impact on their engagement [32,33].

The moderating role of PSC is evidenced in both Malaysia and Australia. For example, the negative impact of detrimental working conditions, such as high emotional demands, on workers’ psychological and emotional health is reduced in high PSC environments [30,32,34]. In organizations with high levels of PSC, a variety of relevant resources are developed for workers to help manage the potentially health damaging working conditions (e.g., high job demands; [35]). The level of PSC also signals to workers the availability and accessibility of the resources [35]. In high PSC organizations, resources are not only provided, their availability is also communicated across the different levels of the organization, as well as senior management encouraging workers to access the available resources. As such, workers can utilize the developed resources that workers have been informed about and encouraged to access to help minimize or manage the detrimentally high job demands and the consequent impact upon their health. Conversely, the positive impact of high job demands upon workers’ engagement was enhanced; in that, workers in high PSC environments experienced enhanced engagement in reaction to high demands than in low PSC environments [32]. It is under conditions of high stress, such as high job demands that resources become salient to the worker [32]. As such, with the increasing levels of PSC in the organization, workers can make use of the now available resources to turn their stressful working conditions into interesting and challenging situations that provide opportunities for self-growth [11]. These opportunities to develop and test skills as part of self-growth provide the correct circumstances for workers to enhance their engagement in their role, consistent with the theory of flow [36]. While there is currently a lack of research exploring the moderating role of PSC among workers in Sweden, it is expected that, consistent with Australia and Malaysia, PSC will reduce the negative impact of high emotional demands on workers’ emotional exhaustion and enhance the positive impact on their engagement.

### 1.3. Source of Emotional Demands

The findings described above have largely investigated emotional demands that arise from interactions with end-users (e.g., clients and patients), or have been unspecified in terms of the source of the emotional demands. However, service employees have different work-related roles both as external service providers (interacting with clients) and as members of the organization (interacting with co-workers; [37]). As such, it seems plausible that these different relational contexts can be distinct sources of emotional demands. Although some studies have suggested that emotional demands in service workers can also arise from work relations [38], most of the research on emotional demands has focused on clients’ relations and therefore knowledge of different sources of emotional demands at work remains largely unexplored. Moreover, some evidence suggests this may also be the case for human service managers, for which problems related to employee groups or individuals, characterized by conflicts, distrust, disciplinary actions, for example, can further add to the emotional demands of the job [39]. However, research on potential differences in emotional demands between service workers and service managers is largely missing.

### 1.4. Aims

The aim of this study is to explore whether different sources of emotional demands at work relate differently to employees’ psychological ill-health and motivation. As the literature shows, emotional demands are an inevitable feature in service work. Furthermore, studies show that emotional demands contribute to psychological strain, but at the same time may have a positive impact on work-related outcomes such as work engagement, especially under conditions with high resources. However, studies have largely investigated emotional demands that arise from interactions with end-users (e.g., clients and patients), or have been unspecified in terms of the source of the emotional demands. The present study uses a new measure of emotional demands that differentiates between sources of emotional demands at work, namely clients (“client” is used here and throughout the paper to refer to any person who interacts with an employee, for example, patients, customers, passengers, guest, etc.) and other employees (superiors, subordinates, and colleagues). Regarding outcomes, we chose to focus on emotional exhaustion based on the match hypotheses of the demand induced strain compensation model [12], according to which demands (e.g., emotional) are likely to have the greatest impact on outcomes in the same domain (e.g., emotional exhaustion). Furthermore, we focus on work engagement as a measure of work-related well-being. Work engagement can be defined as an affective-motivational state, which is characterized by vigor, dedication, and absorption with one’s job [40].

As suggested by previous literature, the organizational context often has an impact on work conditions (e.g., emotional demands) and on psychological health outcomes. Therefore, interactions will be analyzed where psychosocial safety climate will be a moderating variable of the relations between emotional demands and the outcomes.

## 2. Materials and Methods

### 2.1. Participants and Procedures

A cross-sectional survey at the request of the research group was conducted by Statistics Sweden (SCB) during autumn 2018 by postal mail including an information letter, a paper version together with a stamped return envelope, and a personal link to a web questionnaire. Non-respondents received up to two reminders, the last of these included new paper questionnaires and return envelopes. In total, we received 3642 responses, which based on the illegible population of 11,556 randomly selected persons gives a gross response rate of 30.9%. For the present study, the inclusion criteria were 25–65 years old employees stating to have direct contact to clients, patients, pupils etc., and in a position of having colleagues and a supervisor. In total, replies from 2742 service workers were included. Of these, 1237 were men (43.8%) and 1587 were women (56.2%). Age mean was 47.72 (SD = 10.76), and ranged between 25 and 65 years. Regarding their occupation, 1891 were employees with no managerial position (67.2%), 450 were managers with staff responsibility (16%), and 472 were managers without staff responsibility (16.8%). The study was approved by the Ethical Board of Southern Sweden (Dnr. 2018-293). Informed consent was obtained from all participants.

### 2.2. Measures

Emotional demands: Emotional demands were measured by four items, rated on a scale from 0 (Never/Almost Never) to 100 (Always). Items asked participants to rate how often they experience emotionally stressful situations in relation to patients, clients, students, customers etc. (Item 1), superiors (Item 2), colleagues (Item 3), and subordinates (Item 4). Items 2, 3, and 4 were averaged to obtain a total score of emotional demands from work relations.

Psychosocial Safety Climate: The present study used the short PSC version (PSC-4: [30,41,42]), which includes four of the 12 items from the full-length version. Participants rated each item on a 5-point scale, ranging from 1 (strongly disagree) to 5 (strongly agree). An item example is, “Senior management show support for stress prevention through involvement and commitment.”

Emotional Exhaustion: Emotional exhaustion was measured by 3 items from the 4-item burnout scale from COPSOQ III [43], in its Swedish version [44]: “How often have you felt worn out? How often have you been physically exhausted? How often have you been emotionally exhausted?” These items refer to the last 4 weeks and are measured on a 5-point Likert scale (All the time (100); A large part of the time (75); Part of the time (50); A small part of the time (25); (0).

Work Engagement: The Utrecht Work Engagement Scale (UWES) measures three dimensions of work engagement: vigor, dedication, and absorption [40]. The present study used the short version with three items from the COPSOQ III [43,44], rated on a 5-point Likert scale from 0 (Never/Almost Never) to 100 (Always). An item example is, “I am enthusiastic about my work.”

### 2.3. Data Analyses

The data were analyzed using SPSS (v24) (IBM, Armonk, NY, USA) and Mplus software (v8) (Muthén & Muthén, Los Angeles, CA, USA). Descriptive statistics such as frequency, percentage, mean, and standard deviation were used to depict the characteristics of the sample and the variables in study. To compare differences in measured variables (emotional exhaustion, work engagement, emotional demands, and psychosocial safety climate) between groups (employees, managers with staff responsibility, and managers without staff responsibility), analyses of variance (ANOVA) were used. To test the hypotheses and examine model fit and relationship between the variables, a path analysis with maximum likelihood estimation with robust standard errors (MLR) was used in Mplus (Muthén & Muthén, Los Angeles, CA, USA). Specifically, three separate path analysis for the three different groups were calculated, in which emotional demands and the interaction terms between emotional demands and psychosocial safety climate were regressed on emotional exhaustion and work engagement. The conceptual model is depicted in Figure 1.

## 3. Results

### 3.1. Descriptive Statistics and Analysis of Variance

Table 1 reports the results of the ANOVA between the three groups on the variables in the study. Results of the LSD post hoc test, which compares means between the three groups two by two, suggest that employees reported more emotional exhaustion, less work engagement, and the lowest levels of PSC, compared to the other groups. Managers without staff responsibility reported the lowest levels of emotional demands arising from client’s relations, while managers with staff responsibility reported the highest levels of emotional demands from work relations.

### 3.2. Path Analysis

Figure 2, Figure 3 and Figure 4 report the results of the path model with moderation. For employees, we found direct effects of emotional demands from both sources on emotional exhaustion. The more emotional demands respondents experienced the higher emotional exhaustion they reported, and the strongest association was seen for emotional demands arising from work relations. For managers with staff responsibility there was a positive direct effect of emotional demands from relations with clients on emotional exhaustion, while for managers without staff responsibility there was a direct effect of emotional demands from work relations on emotional exhaustion.

Regarding direct effects on work engagement results showed that, for managers with staff responsibility, emotional demands from clients predicted lower work engagement, while the opposite result was found for managers without staff responsibility. Results for employees were not significant.

Finally, results showed two significant interactions: (a) for employees, PSC moderated the association between emotional demands from work relations and emotional exhaustion; (b) for managers without staff responsibility, PSC moderated the association between emotional demands from work relations and emotional exhaustion.

### 3.3. Simple Slopes Analysis

Using simple slopes analysis, we found that with increasing levels of PSC the positive association between emotional demands and exhaustion reduced for all three groups (Table 2).

For example, under conditions of low PSC there was a significant association between emotional demands and exhaustion for all three groups, which reduced with medium levels of PSC and became non-significant in conditions of high PSC (only for managers).

## 4. Discussion

The purpose of the present study was to explore the association between emotional demands from different relational sources at work and emotional exhaustion and engagement in service workers. Although the relation between emotional demands and indicators of psychological strain and other job-related outcomes has been broadly investigated, e.g., [14], previous studies largely focused on emotional demands arising from interactions with end-users (e.g., clients and patients), or have been unspecified in terms of the source of the emotional demands.

Consistent with previous research, it was found that high emotional demands led to increased symptoms of emotional exhaustion [45,46,47]. Specifically, it was found that, among employees, emotional demands from both clients and especially from work relations contributed to feelings of emotional exhaustion. Emotional demands erode workers’ emotional health because the increased effort associated with the appraisal of demands and coping with them results in strain (e.g., anxiety and fatigue), which over time can lead to employees feeling exhausted and worn out.

This link between emotional demands from “both” work relations and clients onto emotional exhaustion was not found for managers. For managers with staff responsibility, only emotional demands from relations with clients significantly predicted emotional exhaustion, while for managers without staff responsibility only emotional demands from work relations predicted emotional exhaustion. Some previous studies also failed to find an association between emotional demands and psychological strain, e.g., [21,22,23]. For example, in a sample of human resource professionals, the interpersonal demands of frequency, duration, and intensity of contact with clients were not significant predictors of emotional exhaustion [21]. Moreover, in a longitudinal study [48], only hindrance demands positively predicted strain at three months, whereas challenge stressors did not, in contrast to the findings of cross-sectional research. The authors of this study suggest that it is likely that previous findings from cross-sectional data, demonstrating a positive relationship between challenge stressors and psychological strain, could be either a result of the strain caused by challenge stressors being short-lived, or of stressed employees overreporting exposure to all stressors. In the present study, employees were significantly more emotionally exhausted than managers, which could explain these findings.

Another possible explanation for our findings could be that, emotional demands perceived as part of the role (e.g., in the case of managers with staff responsibility, emotional demands from work relations) are less straining, but emotional demands that might not be perceived as part of the role (e.g., emotional demands from clients) contribute to more strain and emotional exhaustion. The idea that job demands can have different consequences for job-related outcomes can be explained by an extension of the JD-R model that distinguishes between challenge job demands and hindrance job demands [46,49]. The general assumption of this dimensional framework is that demands that tend to be viewed as barriers to goal accomplishment and are therefore considered inhibitory to personal growth can be considered as hindrance demands [49]. In contrast, job demands that can create an opportunity for personal growth and development can be considered as challenge demands [49]. Relating this to our findings, it is possible that emotional demands that are not perceived as part of the role work as hindrance demands, and therefore contribute to strain and emotional exhaustion, while this is not always the case for emotional demands that might be perceived as part of the role. For example, for a manager with staff responsibility, emotionally demanding situations related to interactions with co-workers, such as conflicts or disciplinary actions, are somehow expected and part of task execution, and therefore may not require additional cognitive, emotional or behavioral coping resources that would otherwise drain energy and lead to emotional exhaustion. In a similar line, a recent study found that some social stressors at work, such as coordination requirements, mediation between colleagues, or enforcing of decisions can be classified as challenge stressors, promoting opportunity for self-efficacy, although at the same time evoking strain [50].

Another possible explanation, which could help understand the different results found for employees and managers, is that managers might have more job resources available that can be tapped to mitigate the potential negative effects of emotional demands on emotional exhaustion, particularly for those demands as seen as part of the job. There is ample evidence supporting the role of job resources as buffers of the negative impact of emotional demands, e.g., [9,12]. In a previous study conducted in Sweden it was found that managers in general have a more favorable work environment and resources when compared to all the other groups [44].

Regarding the impact of emotional demands on work engagement, we found that for managers without staff responsibility, emotional demands from clients predicted higher work engagement, while for managers with staff responsibility, emotional demands from clients predicted lower work engagement. Results for employees, however, were not significant.

Using the challenge–hindrance framework to explain these findings, it is possible that for managers with staff responsibility, dealing with stressful situations from clients, may be appraised more as hindrance demands, as these may not be perceived as part of the role and therefore they may less likely to invest their time and energy in responding to these demands as they think their efforts will not result in success, but rather more exhaustion [47]. In the same line, managers without staff responsibility may have more client interaction as part of their role and therefore more opportunities to develop appropriate resources to deal with emotionally demanding situations from this source; instead of leading to strain, these interactions may provide opportunities for growth and development, feelings of self-efficacy and optimism, as the employee accumulates knowledge that challenging problems can be solved, which then leads to more work motivation and engagement. A previous meta-analytic study also showed that demands which employees tend to appraise as hindrances were negatively associated with engagement, and demands that employees tend to appraise as challenges were positively associated with engagement [47].

The finding that emotional demands from clients were not related to engagement for employees was unexpected. However, other potential moderating variables not measured in the current study might explain this result. In a previous study, for example, a non-significant relationship between emotional demands and work engagement was found only for employees with low levels of self-efficacy, while for employees with higher levels of self-efficacy this relationship was positive [11].

Finally, we also found that PSC moderate the path from emotional demands to exhaustion for all workers independently of position. Specifically, organizations with higher levels of PSC have workers reporting a weaker detrimental association between their emotional demands and emotional exhaustion. This moderating role of PSC between demands and health outcomes for workers is a recognized finding in the literature, e.g., [32,33,35]. PSC acts as a macro-level resource as well as a safety signal to combat the health eroding effects of high job demands in the workplace [35]. That is, workers will be provided with and directed to resources in their workplace to help combat these often-unavoidable demands in the service sector. High PSC also signals to the workers that senior management prioritizes their health and safety, providing workers with a safe space to voice concerns over working conditions (e.g., high emotional demands) as well as systems to manage the high emotional demands (e.g., providing sufficient recovery time).

Our results did not support the hypothesis that PSC would also moderate the effects of emotional demands on work engagement. Previous studies found moderating effects of PSC on work engagement, by decreasing the negative impact of general job demands [32] and bullying/harassment [51]. However, no previous studies explored whether PSC moderated the impact of emotional demands on work engagement, so results cannot be directly compared. One possible explanation is that we only found one significant negative direct effect of emotional demands (from clients) on work engagement, but close to being non-significant.

### 4.1. Limitations and Plans for Future Research

The present study has several limitations that should be taken into account when interpreting the results. First, the relatively low response rate (30.9%) can limit the generalizability of the findings. Furthermore, the data are cross-sectional which does not allow to establish causal relations between the variables. This could be overcome by designing longitudinal studies where variables are measured at different time points. Additionally, data was collected using self-reported measures and therefore subject to common method bias. Improvements in the design could be to use a combination of self-reports with third party reports of employee psychological health, (e.g., clinical assessment) and/or objective measurements (e.g., physiological assessment); or to use an aggregated measure of psychosocial safety climate by sampling several individuals in the same unit to obtain an organization-level climate measure, e.g., [34]. Moreover, our measure of emotional demands relies on individual perception, and therefore may depend on one’s own psychological state and coping. Previous studies have raised concerns about bias in studies examining perceived emotional demands, e.g., [18]. Because this is the first study to employ this measure of emotional demands, results need to be replicated in subsequent research. In the same line, perceptions of emotional demands at work as more in-role or extra-role may also vary between individuals, and so future studies should take individual attributions into account rather than establishing these categorizations a priori. Future studies could also control for the potential effects of sociodemographic characteristics in the model, such as age. Previous studies have explored how emotional demands interact with other job demands (e.g., [52] and resources e.g., [29]) in predicting well-being and work engagement. This body of research suggests that when other types of demands are absent or low, or when appropriate resources are provided, emotional demands increase employees’ engagement with their role, or in other words, can be perceived as a positive “challenge”. Therefore, future studies could explore these interactions with the new measure of emotional demands.

### 4.2. Research Conclusions and Implications

Results from this study offer several theoretical and practical implications. Theoretically, this is the first study to our knowledge to have differentiated between sources of emotional demands, and therefore offers a new avenue of research for future studies looking at emotional demands at work. Moreover, this study offers support to the distinction between challenge and hindrance demands, e.g., [47], and adds to the literature by suggesting that whether emotional demands are perceived as challenge or hindrance may depend on the relational source and, consequently, whether they are seen part of the role. The finding that challenge and hindrance emotional demands were both positively related to emotional exhaustion, while differently related to engagement suggests that these are not opposites poles of the same continuum as has been previously suggested [53]. Rather, it suggests that individuals who may at some point feel exhausted can at the same time be willing to invest in their work and feel engaged when they are confronted with demands, especially if these are perceived as important and meaningful. Finally, this study adds to the body of evidence supporting PSC as a significant moderator within the broad JD-R framework, protecting workers from the negative outcomes of job demands, e.g., [32].

Several important practical implications may be drawn from this study. First, emotional demands were found to be related to service workers’ engagement, suggesting that they can benefit from coping successfully with emotional demands perceived as part of their role. However, frequent exposure to emotional demands is also linked to emotional exhaustion, and therefore it may not be desirable to promote situations that can be a source of emotional demands. Moreover, given that emotional demands are closely tied to work tasks especially for service workers, these can be hard to avoid. Therefore, human resources practices that provide employees with working conditions that allow them to manage emotional demands, for example, by decreasing other types of job demands, e.g., [52], or by increasing job resources, e.g., [29], may promote engagement but not at the expense of their psychological well-being. Similarly, efforts to work on modifying PSC in organizations could increase employees’ resilience to the negative effects of emotional demands on psychological well-being, e.g., [54]. Finally, although we cannot rule out that employees who feel emotionally exhausted simply perceive that they have greater emotional demands, it is also reasonable to suggest that managers could potentially reduce an employee’s emotional exhaustion by limiting the emotional demands the employee must cope with, especially those perceived as hindrances.

## 5. Conclusions

The main goal of the present study was to examine the relation between different sources of emotional demands at work and employee’s well- and ill-being. Up to now, studies have focused on emotional demands that arise from interactions with end-users (e.g., clients and patients), or have been unspecified in terms of the source of the emotional demands. However, service employees have different work-related roles both as external service providers and as members of the organization and thus it seems plausible that these different relational contexts can be distinct sources of emotional demands. Overall, this study suggests that different relational sources of emotional demands at work have a different impact on employees’ emotional exhaustion and engagement. When emotional demands are seen as part of the role, such as dealing with stressful situations with clients, not only their negative impact is significantly reduced but it can also contribute to more motivation and engagement with work. Emotional demands seen as extra-role, such as dealing with stressful situations with a manager or co-workers, seem to act as roadblocks to individual growth and development and therefore can lead to exhausted and disengaged workers. The present study provides support for the challenge–hindrance stressor framework and extend it by differentiating sources of emotional demands. Moreover, by distinguishing the source of the emotional stressor and the occupational role we were able to better understand previous mixed results regarding the association between emotional demands and exhaustion. Strategies that promote a reduction of extra-role emotional demands, and at the same the development of a psychosocial safety climate in the organization, could therefore offer interesting solutions to promote employees’ psychological well-being and motivation at work.

## Figures and Tables

**Figure 1 ijerph-17-07738-f001:**
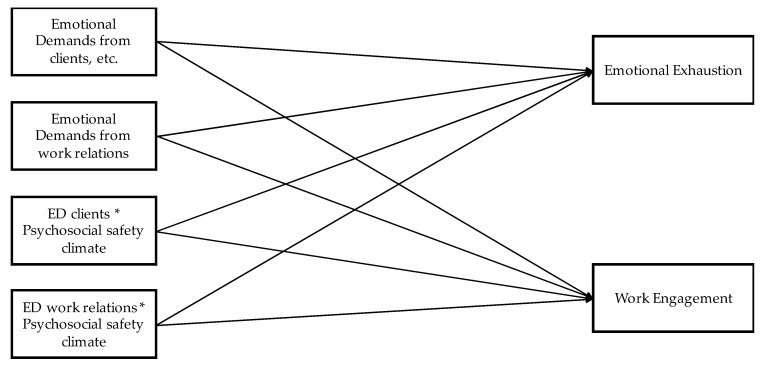
Conceptual model of the path analysis. Note: * = interaction.

**Figure 2 ijerph-17-07738-f002:**
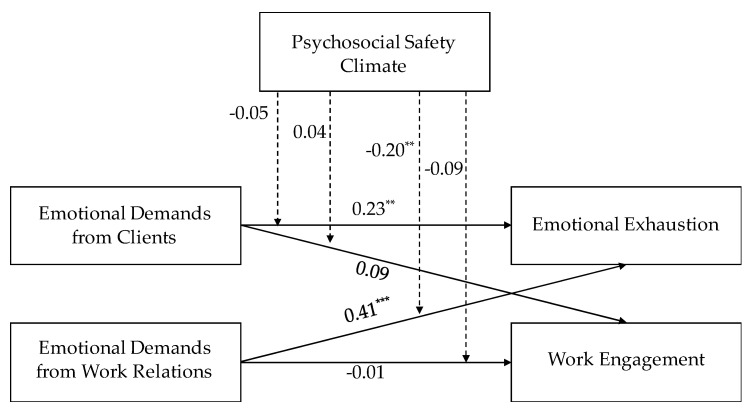
Standardized coefficients of the Path Analysis for employees. ** ≤ 0.01; *** ≤ 0.001.

**Figure 3 ijerph-17-07738-f003:**
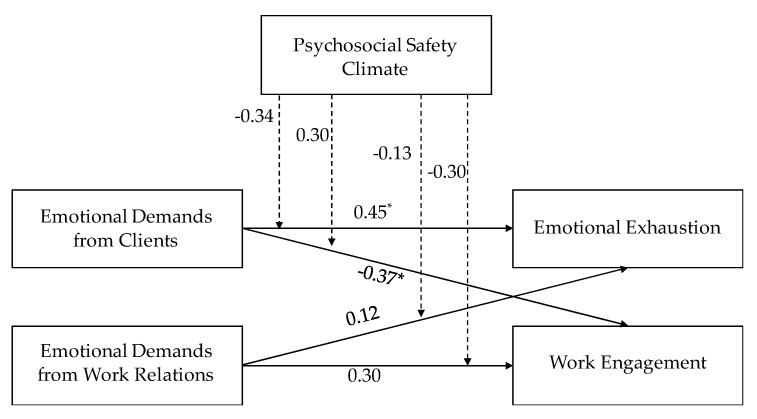
Standardized coefficients of the Path Analysis for managers with staff responsibility. * ≤ 0.05.

**Figure 4 ijerph-17-07738-f004:**
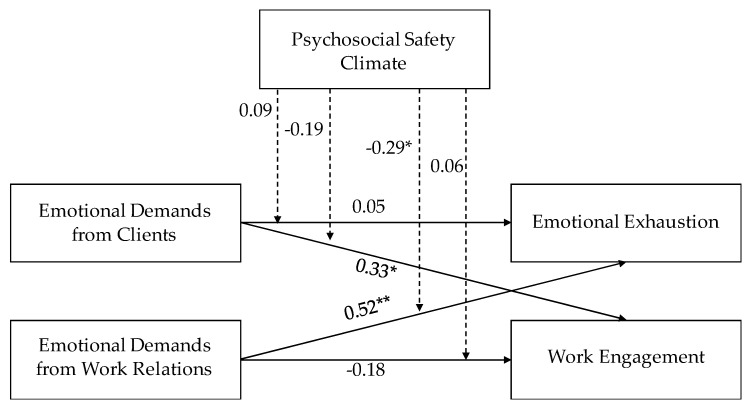
Standardized coefficients of the Path Analysis for managers without staff responsibility. * ≤ 0.05; ** ≤ 0.01.

**Table 1 ijerph-17-07738-t001:** Means, standard deviations, analysis of variance, and significant differences from the post hoc tests between groups on the study’s outcomes.

Variable	Group	M	SD	F	*p*	LSD Post Hoc Test *
Emotional exhaustion	1. Employees	36.02	24.96	3.89	0.01	1 > 2 1 > 3
	2. Managers with staff	33.25	23.39			
	3. Managers without staff	33.34	22.27			
Work engagement	1. Employees	68.22	19.02	40.79	<0.001	2 > 1 2 > 3 3 > 1
	2. Managers with staff	76.37	16.58			
	3. Managers without staff	72.70	17.18			
Emotional demands from clients	1. Employees	43.44	27.71	4.66	0.010	1 > 3 2 > 3
	2. Managers with staff	44.04	25.55			
	3. Managers without staff	39.18	25.47			
Emotional demands from work relations	1. Employees	28.24	20.72	22.57	<0.001	2 > 1 2 > 3
	2. Managers with staff	35.46	19.58			
	3. Managers without staff	28.56	19.90			
Psychosocial safety climate	1. Employees	2.80	0.99	29.13	<0.001	2 > 1 2 > 3 3 > 1
	2. Managers with staff	3.19	0.98			
	3. Managers without staff	2.93	1.02			

Note. * only significant mean differences between the groups are reported here (*p* ≤ 0.05).

**Table 2 ijerph-17-07738-t002:** Simple slopes analysis from emotional demands to emotional exhaustion.

Group	Psychosocial Safety Climate		
	Low	Medium	High
Employees			
Simple slope	0.29	0.22	0.16
Significance	<0.001	<0.001	<0.001
Managers with staff responsibility			
Simple slope	0.24	0.14	0.06
Significance	<0.001	0.005	0.342
Managers without staff responsibility			
Simple slope	0.34	0.23	0.15
Significance	<0.001	<0.001	0.051

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
