# Peer review of "Not All Emotional Demands Are the Same: Emotional Demands from Clients’ or Co-Workers’ Relations Have Different Associations with Well-Being in Service Workers"

_ijerph, 2020, doi:10.3390/ijerph17217738_

Round 1

Reviewer 1 Report

The study is of scientific interest providing useful information.  The introduction provides relevant information. It is well written, approaching the issue clearly. The objectives are timely and well planned

It is methodologically correct. The procedure, the selected measuring instruments and the analyzes performed are correct.

The results section clearly shows the data obtained.

In the discussion, comparisons of the data obtained with those found in other studies are established, proposing answers to the discrepancies found and new avenues of investigation

They clearly state the conclusions, the usefulness of the data found and their applicability. In addition, indicate the future lines in which it would be necessary to deepen.

In the bibliography, in references 5, 8 and 12 they must correctly write the name of the authors (they appear lowercase when they must appear uppercase)

Author Response

Dear reviewer, 

Thank you for your comments regarding our paper.

Regarding your suggestion, we have now corrected the names of the authors for references 5, 8, and 12. 

Reviewer 2 Report

I enjoyed reading this study which for a complex area, was clearly presented. 

I'm not sure what "industry 4.0" means in line 35 and elsewhere. 

The response rate of 30.9% was low and the final number of responses analysed fell by a further 900 . What was the reson for this?

The final sample was around 20% of those invited to participate. Might this impact on the generalisability of the findings. This should be considered in the Limitations.

In figure 1, I presume * refers to an interaction. This should be indicated under the table.

In table 1 the word "Outcome" is confusing and I suggest replacing it with "Variable". 

The body of the discussion is a little long and should be shortened slightly. 

A limitation that should probably be mentioned is that the study didn't investigate reasons for the different impact of PSC and work relations on the outcome measures, although they did consider these in the discussion. 

Was any consideration given to including age in the path analysis becuase it way be that lder more experiened managers and other employees might react differently to their younger counterparts?

This paper most definitely requires a statistical review. 

Author Response

Dear reviewer, 

Thank you for your comments and suggestions regarding our paper. Below we specify how we addressed them.

1. I'm not sure what "industry 4.0" means in line 35 and elsewhere. 

For clarity, we decided to replace the term "industry 4.0" and re-wrote both sentences on Lines 35 and 83. 

2. The response rate of 30.9% was low and the final number of responses analysed fell by a further 900 . What was the reson for this?

The final sample was around 20% of those invited to participate. Might this impact on the generalisability of the findings. This should be considered in the Limitations.

The sample used in this study is a national random sample across many workplaces which, in general, has lower response rates when compared to samples from a single workplace. 

Also, the response rate is comparable to previous studies using national random samples. For example, 7.3% in a Canadian population study (Ramkissoon A, Smith P, Oudyk J. Dissecting the effect of workplace exposures on workers' rating of psychological health and safety. Am J Ind Med 2019;62:412e21.), and 48.4% in a Danish population study (Clausen, T., Madsen, I. E., Christensen, K. B., Bjorner, J. B., Poulsen, O. M., Maltesen, T., ... & Rugulies, R. (2019). The Danish psychosocial work environment questionnaire (DPQ): development, content, reliability and validity. Scandinavian Journal of Work, Environment & Health45(4), 356-369.)

The further reduction of the sample size by 900 is related to the inclusion criteria for this study (25-65 years old employees stating to have direct contact to clients, patients, pupils etc., and in a position of having colleagues and a supervisor) which led to the exclusion of those subjects.

According to the reviewer suggestion, we included the low response rate in the limitations of the study (line 422).

3. In figure 1, I presume * refers to an interaction. This should be indicated under the table.

This was added to the figure.

4. In table 1 the word "Outcome" is confusing and I suggest replacing it with "Variable". 

This was replaced.

5. The body of the discussion is a little long and should be shortened slightly. 

The authors excluded and rephrased some parts of the discussion in order to shorten it (line 280, 324, 339, 358). Given the complexity of this study, with multiples variables and outcomes, comparisons between several groups and the novelty of the topic, the authors were concerned that further shortening the discussion could comprise its clarity. 

6. A limitation that should probably be mentioned is that the study didn't investigate reasons for the different impact of PSC and work relations on the outcome measures, although they did consider these in the discussion. 

The authors agree that there could be different reasons for the different impact of PSC and work relations on the outcome measures that could be further explored. However, this was not an aim of the present study, and therefore the authors don't see this as a limitation, but perhaps an area for further study.

7. Was any consideration given to including age in the path analysis becuase it way be that lder more experiened managers and other employees might react differently to their younger counterparts?

This is an interesting point. However, no significant correlation was found between age and emotional demand variables and only weak correlations with exhaustion and engagement. We added a sentence in the discussion recommending the inclusion of age and other sociodemographic characteristics in future studies (line 437).

8. This paper most definitely requires a statistical review. 

The authors are not sure what the reviewer mean by this. There were no comments from the other reviewers regarding the statistics so we believe the statistical procedures used are appropriate. 

Reviewer 3 Report

Good research work, just two notes:
1. In the abstract the acronym PSC is not previously described in the text.
It is not possible to know what the acronym corresponds to (line 25).
2. In the introduction, the term "psychological-quantitative demands"
is not defined, which would be more clarifying (line 60).

Author Response

Dear reviewer, 

Thank you for your comments regarding our paper. 

According to your suggestions, we: 1. described the acronym PSC in the abstract (line 25), and 2. provided an example of psychological-quantitative demands" to clarify it (line 66).

Reviewer 4 Report

I like the article very much, but I would suggest rewriting the last parts: highlighting three areas:
1. research conclusions
2. conclusions
3. discussion and plans for further research

Author Response

Dear reviewer, 

Thank you for your comments regarding our paper. 

However, we are not sure how to address this point. For example, we are not sure what is the difference between 'research conclusions' and 'conclusions'. We renamed the section 'Implications' to 'Research conclusions and implications' as it covers the conclusions from the study as well. Also, we addressed plans for future research together with the limitations of the study. 

Reviewer 5 Report

The present paper aimed at investigating the relationship between emotional demands from relational sources and service workers’ exhaustion and engagement in a sample of 2742 service workers. Results showed emotional demands to be positively associated with emotional exhaustion, particularly if dealing with clients was not an integrated part of the professional role. Further, there was opposite findings in the relationship between emotional demands and engagement for managers with staff responsibility with respect to managers without staff responsibility. Finally, emotional demands showed strongest effect on emotional exhaustion when psychosocial safety climate was low, with a moderating effect.

Manuscript is well structured. It is clear describing results found in a simple and descriptive way. However, some considerations that would help to increase the quality of the said work, could be taken into account.

  1. Abstract:
    1. Authors should add the sample size of the study sample.
    2. The term “PSC” (psychosocial safety climate) has not been previously defined. Please check.

  1. My main concern is related to the instrument used for the assessment of Emotional Exhaustion. As stated in Methods, Authors used 3 items from the burnout scale of COPSOQ III, besides this scale was not specifically validated to assess Emotional Exhaustion. Conversely, the use of a specific validated scale, as the 9-item Emotional Exhaustion (EE) scale of the Maslach Burnout Inventory Scales, might have been more appropriate. Moreover, the assessment of Emotional Exhaustion with only 3 items may have affected the strength of the results. Authors should declare this point when discussing the limitations of the present study.

  1. Results showed moderating effects of psychosocial safety climate on the relationship between emotional demands and emotional exhaustion for employers and managers without staff responsibility. Particularly, emotional demands had the strongest effect on emotional exhaustion when psychosocial safety climate was lower, thus suggesting the importance of the improvement of specific policies, practice and organizational resource targeted to workers’ well-being. This to me is a relevant point of this study. Similar results were found in healthcare workers, a work category characterized by high emotional demands and exhaustion, leading to work-related posttraumatic stress and burnout. For example, a recent systematic review on healthcare workers facing Coronavirus outbreaks, found perceived safety of the working environment and a good work organization, as factors which seem to protect healthcare workers to the development of work-related posttraumatic stress, as well as a clear communication of directives and supervisors’ and colleagues’ support (see doi: 10.1016/j.psychres.2020.113312). Psychosocial safety climate seems to be a primary goal of work well-being intervention strategies, particularly during the current COVID-19 pandemic emergency. It could be useful briefly comment this point when discussing their results.

Author Response

Dear reviewer, 

Thank you for your comments regarding our paper. 

1. According to the reviewer suggestion we added the sample size (line 20) to the abstract and defined PSC (line 25). 

2. The scale used to measure Emotional Exhaustion has been previously used in the validation study of the Swedish version of The Copenhagen Psychosocial Questionnaire, COPSOQ III (Berthelsen, H., Westerlund, H., Bergström, G., & Burr, H. (2020). Validation of the Copenhagen Psychosocial Questionnaire Version III and establishment of benchmarks for psychosocial risk management in Sweden. International Journal of Environmental Research and Public Health17(9), 3179). The Swedish version of the COPSOQ III maintains 3 items from the original 4-item Burnout scale, and showed reliability and construct validity.  This decision was made based on perceived relevance to the Swedish context, cognitive interviews, pilot tests and dialogue with stakeholders. 

The original 4-item burnout scale of the COPSOQ III was based on the Personal Burnout subscale from the The Copenhagen Burnout Inventory (Pejtersen JH, Kristensen TS, Borg V, Bjorner JB. The second version of the Copenhagen Psychosocial Questionnaire. Scand J Public Health. 2010 Feb;38(3 Suppl):8-24). 

The authors of The Copenhagen Burnout Inventory highlight several concerns regarding the use of the Maslach Burnout Inventory Scale (see validation paper above), which led to the development of this new burnout scale. 

One of the main issues in using scales with few items is related to low reliability, which was not the case in the validation study of the COPSOQ-III Swedish version. 

Thus, the authors believe that the use of the Emotional Exhaustion scale was theoretically and methodologically appropriate. 

3. The authors included more information about the moderating role of PSC in the discussion (line 402). In the conclusion section, the authors mention that "Strategies that promote a reduction of extra-role emotional demands, and at the same the development of a psychosocial safety climate in the organization, could therefore offer interesting solutions to promote employees’ psychological well-being and motivation at work", in line with the reviewer's suggestion (line 497).